# Human Motion State Recognition Based on Flexible, Wearable Capacitive Pressure Sensors

**DOI:** 10.3390/mi12101219

**Published:** 2021-10-06

**Authors:** Qingyang Yu, Peng Zhang, Yucheng Chen

**Affiliations:** 1College of Control Science and Engineering, China University of Petroleum, Qingdao 266580, China; 2Key Laboratory for Robot Intelligent Technology of Shandong Province, Shandong University of Science and Technology, Qingdao 266590, China; pengzhang@sdust.edu.cn (P.Z.); sdustchenyucheng@163.com (Y.C.)

**Keywords:** flexible capacitive pressure sensor, microstructured electrode, multi-walled carbon nanotube, barium titanate, back propagation neural network, human motion state recognition

## Abstract

Human motion state recognition technology based on flexible, wearable sensor devices has been widely applied in the fields of human–computer interaction and health monitoring. In this study, a new type of flexible capacitive pressure sensor is designed and applied to the recognition of human motion state. The electrode layers use multi-walled carbon nanotubes (MWCNTs) as conductive materials, and polydimethylsiloxane (PDMS) with microstructures is embedded in the surface as a flexible substrate. A composite film of barium titanate (BaTiO_3_) with a high dielectric constant and low dielectric loss and PDMS is used as the intermediate dielectric layer. The sensor has the advantages of high sensitivity (2.39 kPa^−1^), wide pressure range (0–120 kPa), low pressure resolution (6.8 Pa), fast response time (16 ms), fast recovery time (8 ms), lower hysteresis, and stability. The human body motion state recognition system is designed based on a multi-layer back propagation neural network, which can collect, process, and recognize the sensor signals of different motion states (sitting, standing, walking, and running). The results indicate that the overall recognition rate of the system for the human motion state reaches 94%. This proves the feasibility of the human motion state recognition system based on the flexible wearable sensor. Furthermore, the system has high application potential in the field of wearable motion detection.

## 1. Introduction

Human motion state recognition technology has been widely applied in different fields such as medical treatment [1,2], human–computer interaction [3,4], and home life [5,6]. The current research regarding human motion detection can typically be divided into two modes. The first mode is the human motion state recognition based on video and image processing technology. For example, Ma et al. [7] of Nanjing University of Posts and Telecommunications proposed an architecture of time-varying long short-term memory recurrent neural networks (TV-LSTMs) for human action recognition, which adopted convolutional neural networks (CNNs) as the feature generator, and obtained good recognition results. Bilen et al. [8] of the University of Edinburgh combined motion pictures and dynamic optical flow with CNNs and summarized the video as an image, which can realize end-to-end video action recognition. Although this vision-based recognition mode has the advantages of strong real-time performance, simple data acquisition, and good continuity, most of the studies focus on algorithm design with a long system development cycle. In addition, slight human movements are difficult to capture using hardware such as cameras. Natural factors such as ambient lighting and object occlusion can also affect the recognition effect to a certain extent. Moreover, the range of human movement is limited by the position of the camera, which can easily invade personal privacy. The second mode is the human motion state recognition based on wearable devices. For instance, Chamroukhi et al. [9] of the University of Toulon in France used an MTx-Xbus inertial tracker on the chest, right thigh, and left ankle of the experimenter to collect data. The problem of motion recognition was solved by segmenting the multi-dimensional time series of acceleration data. Moreover, Ronao et al. [10] of Yonsei University in South Korea adopted smartphones as sensors and a deep CNN as the recognition model and used the unchanging characteristics of the one-dimensional time input signals to realize the recognition of human motion state. In the above-mentioned models, the motion data are collected using wearable rigid sensors and thus have a flexible application range. However, the way of wearing hardware devices may restrict people’s joint activities to a certain extent, and the human–computer interaction experience is not sufficiently effective. Therefore, a compact, lightweight, and wearable sensor with superior performance is required to collect human motion signals. With the rapid development of flexible electronic technology, the emergence of flexible sensors provides new ideas for the detection of human motion state. Unlike the traditional rigid sensors, flexible sensors can adapt to complex detection environments and are small and lightweight. Thus, such sensors can adapt well to the requirements of human motion monitoring and not cause great obstacles to the activities of the human body [11,12,13,14]. Although there have been reports on the application of flexible sensors in the field of motion state recognition, the recognition effect is still limited by key factors such as the sensitivity and stability of the sensor. Hence, it is of certain significance and value to design a flexible sensor with superior performance that can be applied to the detection of human motion state. There are many types of flexible pressure sensors, and their sensing mechanisms are mainly of piezoresistive sensors [15,16], piezoelectric sensors [17,18], and capacitive sensors [19,20]. Piezoelectric sensors have the advantages of high sensitivity and fast response due to their unique sensing principles. However, they cannot effectively measure static pressure [21]. Piezoresistive sensors have the advantages of easy data collection, simple production, and stable performance. However, the sensing materials can be easily affected by environmental factors. Piezoresistive sensors can be strongly affected by the viscoelasticity of flexible materials and are prone to large hysteresis, thereby affecting the real-time detection effect [22]. In comparison, capacitive sensors have low power consumption, high sensitivity, low hysteresis, fast response, and good perception of small stress; thus, they can meet the actual requirements of human motion detection [23]. The capacitance is calculated as follows [24]:(1)$$C=\frac{\varepsilon_{0}\varepsilon_{r}A}{d},$$
where $\varepsilon_{0}$ is the vacuum dielectric constant, $\varepsilon_{r}$ is the relative dielectric constant of the dielectric layer, A is the relative area of the two electrode layers, and d is the distance between the two electrode layers. Among them, variable parameters A, d, and $\varepsilon_{r}$ affect the sensing performance such as sensitivity of the sensor. The sensitivity of a capacitive pressure sensor is as follows [25]:(2)$$S=\frac{\delta \left((C-C_{0})/C_{0}\right)}{\delta P},$$
where S is the sensor sensitivity, C is the capacitance value output by the sensor after pressure is applied, $C_{0}$ is the initial capacitance value of the sensor, and P is the change in pressure received by the sensor. Substituting Equation (1) into Equation (2), we can obtain
(3)$$S=\frac{\delta \left(\frac{\varepsilon_{r}Ad_{0}}{\varepsilon_{r0}A_{0}d}-1\right)}{P},$$
where $\varepsilon_{r0}$, $A_{0}$, and $d_{0}$, respectively, represent the relative dielectric constant of the dielectric layer, the relative area of the electrode plates, and the distance between the two electrodes in the initial state of the sensor; $\varepsilon_{r}$, A, and d, respectively, represent the corresponding parameters when pressure, P, is applied to the sensor. As shown in Equation (3), to increase the sensitivity of the sensor, under the same pressure applied, $\varepsilon_{r}$ and A must be maximized as much as possible and d must be reduced. Currently, the main method to improve the performance of capacitive sensors is to construct microstructures in the dielectric layer and the electrode layer. For example, Bao et al. [26] from Stanford University proposed a dielectric layer integrated with a pyramid-shaped microstructure. The capacitive flexible pressure sensor composed of this dielectric layer had a sensitivity of 0.55 kPa^−1^ in a pressure range of 0–0.2 kPa. The pressure detection limit was as low as 3 Pa, and the response time was less than 1 s. Furthermore, Wei Li et al. [27] from Tiangong University proposed a flexible capacitive pressure sensor based on a polydimethylsiloxane (PDMS) dielectric layer with air gaps and high porosity. The sensor had an excellent sensitivity of 1.15 kPa^−1^ (<1 kPa) in a pressure detection range of 0–5 MPa. However, the volume of the sensor based on the foam structure was relatively large, and the wearability of the human motion detection was poor. Zian Zhang et al. [28] from Sun Yat-sen University reported a flexible, interlayered capacitive pressure sensor with micro-cone array electrodes and a porous medium layer. The sensor had high sensitivity (2.51 kPa^−1^), fast response time (84 ms), and a wide working range (>10 kPa). Clearly, further improvement of the performance of capacitive sensors can be realized by both constructing a microstructure in the electrode layer and improving the dielectric constant. Based on the above discussion, the following strategies are adopted in this study to improve the performance of sensors: using a low-cost sandpaper mold, the micro structure is fabricated on the surface of the sensor electrode. The BaTiO_3_ with high dielectric constant and low dielectric loss is doped into PDMS as filler, and the dielectric constant of the dielectric layer is improved. On the other hand, because multi-layer back propagation neural network (BP neural network) training has strong nonlinear mapping ability and high self-learning and adaptive ability, the BP neural network is used to train and identify the collected samples.

In this paper, based on previous work [29], a human motion state recognition system based on wearable flexible capacitive pressure sensor was designed. Firstly, we designed a flexible pressure sensor consisting of MWCNTs/PDMS electrode layers with microstructures embedded on the surface and a BaTiO_3_/PDMS dielectric layer. The addition of BaTiO_3_ effectively improved the dielectric constant of the dielectric layer and the sensitivity of the sensor. At the same time, the response characteristics and reliability of the sensor were explored using various mechanical deformations and a hot box test. Subsequently, the data acquisition, denoising, sample division, and feature extraction of the corresponding sensing signals (respiration signals and signals of elbow joint and knee joint bending) of volunteers in four motion states (sitting, standing, walking, and running) were completed. Finally, using BP neural network training, a data model of human motion state recognition was established, which realized the recognition of the motion states of sitting, standing, walking, and running. The sensor can effectively monitor human physiological activity information, such as breathing signals and the state of joint motion, and can significantly contribute to the field of wearable motion detection.

## 2. Experimental Section

### 2.1. Preparation of Materials

The MWCNTs were purchased from Nanjing Xanano Materials Tech Co., Ltd., Nanjing, China with 95% purity, 10–30 m length, and 2.0 wt% carboxyl content. The MWCNTs were dispersed in anhydrous ethanol (AET) and stirred manually for 10 min. Subsequently, high-frequency ultrasound was used for 30 min to further disperse the MWCNTs to reduce the occurrence of agglomeration of the MWCNTs in the dispersion. Then, MWCNTs/AET dispersion with a mass fraction of 1 wt% was obtained. The dispersion solution was kept static for 12 h. Subsequently, the bottom precipitation was removed, and the supernatant was retained for further experiments. PDMS (Sylgard 184 model) was purchased from Dow Corning. The PDMS main agent and curing agent were mixed at a ratio of 15:1 and stirred for 30 min, and then they were processed under vacuum for 30 min to remove air bubbles in the mixed solution for later use. Eighty-mesh sandpaper (German Warrior brand) was purchased from a supermarket, cut, and affixed flat on a glass plate. BaTiO_3_ with a particle diameter of 500 nm and content of 99.5% was purchased from Beijing Shenghe Haoyuan Technology Co., Ltd. BaTiO_3_/PDMS-mixed solutions with a mass percentage of 0%, 10%, 20%, and 30% were prepared. The solutions were stirred for 2 h to remove air bubbles. Adhesive polyimide (PI) tape (50 µm thickness) was purchased from 3M Company.

### 2.2. Fabrication of Microstructured Electrode Layer

Based on the previous work [29], the preparation method of the microstructured electrode is shown in Figure 1a. First, the MWCNTs solution was sprayed onto the surface of the 80-mesh sandpaper flatly attached to the glass plate. After heating and drying, a layer of dry MWCNTs was deposited on the surface of the sandpaper. The PDMS solution was poured on the surface, and the glue was homogenized by spin coating for 40 s at a speed of 300 rpm. The PDMS solution penetrated into the MWCNTs layer due to gravity. After heating and curing, the PDMS layer solidified to lock the MWCNTs layer firmly. Thereafter, the PDMS film was peeled off. Thus, the MWCNTs/PDMS electrode layer was obtained. Owing to the uneven microstructure on the surface of the sandpaper, the MWCNTs/PDMS electrode layer displayed a microstructure opposite to the unevenness of the sandpaper surface. Since the MWCNTs layer had a considerably lower thickness than the PDMS layer, this layer hardly affected the elastic properties of the PDMS layer, ensuring that the microstructured electrode had excellent stability, stretchability, and a low manufacturing cost. Figure 2a–c shows the surface morphology of the microstructured electrode layer. It can be seen that the MWCNTs layer is firmly embedded on the surface of the PDMS with microstructure. The close connection between MWCNTs and MWCNTs provides good conductivity for the sensor. In addition, we also characterized conductive MWCNTs layers at different positions of the electrode layer, as shown in Figure 2d–g. Figure 2d,e show that the thickness of the conductive MWCNTs layer at the bottom of the groove on the surface of the electrode layer is 1.094 µm–1.216 µm. Figure 2f,g show that the thickness of the conductive MWCNTs layer on the top of the bulge on the surface of the electrode layer is 3.054 µm–5 µm.

### 2.3. Fabrication and Analysis of Dielectric Layer

The preparation process is shown in Figure 1b. Four BaTiO_3_/PDMS mixed solutions with different contents were spin coated on the smooth glass sheet in sequence for 40 s at a speed of 400 rpm. After the spin coating was completed, the sample was heated at 70 °C for 3 h to cure the composite film. Finally, the composite film was peeled off from the surface of the smooth glass sheet. The surface of four dielectric layers was characterized using a scanning electron microscope (SEM). As shown in Figure 3a–d, the cross-section of the pure PDMS film was the smoothest, and the cross-section of the composite film with 10% BaTiO_3_ addition had sporadic BaTiO_3_ nanoparticles. Furthermore, there were more BaTiO_3_ nanoparticles in the cross-section of the composite film with 20% BaTiO_3_ addition, and BaTiO_3_ particles appeared as small agglomerates. The cross-section of the composite membrane with 30% BaTiO_3_ addition had more BaTiO_3_ particles distributed in the PDMS in the form of small agglomerates. Moreover, large BaTiO_3_ nanoparticle agglomerates were generated. We observed the presence of small aggregates in the cross-section of the composite film added with 30% barium titanate, as shown in Figure 3e. It can be observed that a large amount of barium titanate can accumulate and agglomerate together. If the BaTiO_3_ content was increased beyond 30%, a larger volume of BaTiO_3_ agglomerates appeared in the dielectric layer of the composite material. This considerably affected the stability of the sensor. Therefore, the maximum content of the dielectric layer composite material BaTiO_3_ in this study was set to 30%. We also characterized the surface of barium titanate film with the addition of 10%, 20%, and 30%, as shown in Figure 3f–h. It can be seen that there is no large-area barium titanate aggregation on the surface of the BaTiO_3_/PDMS film. This distribution of barium titanate is helpful to improve the dielectric constant of the dielectric layer.

A three-electrode measurement system (Shanghai Anbiao Electronics Co., Ltd., Shanghai, China) was applied to measure the dielectric constant of the four dielectric layer samples. As shown in Figure 4a,b, the three-electrode measurement system mainly includes the protective electrode, the protected electrode, the common electrode and its accessories, the sample, and the shielding box. The equivalent diagram of a cross-section of the three-electrode measurement system is shown in Figure 4c. The three-electrode measurement system was made of stainless steel. The size parameters of the electrode are as follows: the diameter of the protected electrode was 50 mm, the diameter of the protective electrode was 74 mm, and the gap between the protected electrode and the protective electrode was 2 mm.

The equivalent area of the electrode is calculated as follows:
(4)$$S=\frac{\pi {\left(d_{1}+g\right)}^{2}}{4},$$
where $d_{1}$ is the diameter of the protected electrode, and g is the gap between the protected electrode and the protective electrode. The relative dielectric constant of the tested sample can be calculated as follows:(5)$$\varepsilon_{r}=\frac{Cd}{S\varepsilon_{0}},$$

Substituting Equation (4) into Equation (5), we can obtain:(6)$$\varepsilon_{r}=\frac{4Cd}{\pi {\left(d_{1}+g\right)}^{2}\varepsilon_{0}},$$

We connected the two test terminals of the LCR meter to the protected electrode and the common electrode, respectively, and measured the capacitance value (C) of the sample. The relative dielectric constant ($\varepsilon_{r}$) of the tested sample could be calculated by Formula (6). The relative dielectric constants of the four samples with BaTiO_3_ content of 0%, 10%, 20%, and 30% are shown in Figure 5. The test curve indicates that the dielectric constants of the four samples decreased slightly with the increase in the capacitance measurement frequency. This may be because, when the test frequency is further increased, the change of the dipole vector is slower than that of the applied electric field, resulting in the weakening of polarization and the reduction in dielectric constant of the composites [30]. With the increase in the BaTiO_3_ content, the relative dielectric constant of the dielectric layer also increased. This may be because with the increase in the BaTiO_3_ content, the crosslinking degree of the composite material increases and the polarization enhances. Thus, in this study, 30% (mass fraction) BaTiO_3_ was added to PDMS to prepare composite dielectric layers.

### 2.4. Sensor Packaging

The fabricated MWCNTs/PDMS electrode layer was cut into a strip with a width of 1 cm and length of 4 cm, and the PDMS/BaTiO_3_ dielectric layer was cut into a square with a side length of 1.5 cm. The sensor thus adopted a “sandwich” structure, with the upper and lower layers being the electrode layers and the middle layer being a dielectric layer. The three-layer structure was sealed and bonded using a single-sided adhesive PI film. The schematic diagram of the sensor structure is shown in Figure 6a. A copper foil conductive tape was used to fix the wire on the surface of the electrode layer, making further testing and usage convenient. The actual image of the sensor is shown in Figure 6b. The thickness of the sensor was 1.05 mm, and the effective pressure sensing area was 1 cm × 1 cm. Images of the deformation states (such as bending and distortion) of the flexible sensor are shown in Figure 6c,d. It can be seen that the deformation type of sensor meets the demand of joint movement.

### 2.5. Characterization

The surface morphology of the dielectric layer was characterized using the Apreo SEM (Thermo Scientific, USA). A digital push–pull force gauge (Adburg) and an electric push–pull force gauge (ZQ-990A) (Dongguan Zhiqu Precision Instrument Co., Ltd., Dongguan, China) were used to provide pressure loading for the sensor. A flexible electronic tester (Shanghai Mifang Electronic Technology Co., Ltd., Shanghai, China) was used to provide bending loading for the sensor. An LCR meter (TH2826) (Changzhou Tonghui Electronics Co., Ltd., Changzhou, China) was used to record the changes in electrical signals output by the sensor. The performance test device of the sensor is shown in Figure 7.

## 3. Results and Discussion

### 3.1. Performance of the Pressure Sensor

The sensor was fixed on the base of the digital push–pull force gauge. As pressure was applied, the LCR meter recorded the capacitance changes corresponding to different levels of pressure. Then, the pressure–capacitance change curve was obtained, as shown in Figure 8a. The piecewise linear fitting analysis shows that the sensitivity of the sensor was 2.39 kPa^−1^ in the pressure range 0–0.12 kPa, 0.23 kPa^−1^ in 0.27–2.68 kPa, 0.08 kPa^−1^ in 3.25–9.45 kPa, and 0.02 kPa^−1^ in 10.73–28.73 kPa. Thus, the sensitivity of the sensor gradually decreased as the pressure increased. This may be because the microstructure of the electrode surface is constantly compressed under the action of external forces, approaching the deformation limit. The air groove formed between the electrode layer and the dielectric layer almost disappeared due to compression. When the sensor was under high pressure, the microstructure of the electrode layer was close to the deformation limit and contributed little to the sensitivity. At this time, the dielectric layer was still within its elastic range, and could produce elastic deformation under high pressure, further reducing the distance between the upper and lower electrodes, resulting in a small output response of the sensor. To measure the response time and recovery time of the sensor more efficiently, a 200 mg weight piece was lightly thrown on the surface of the sensor to simulate transient pressure loading. Subsequently, tweezers were used to quickly move the weight piece away from the sensor to simulate pressure unloading. Figure 8b indicates that the response time (Tr) of the sensor was 16 ms when the weight piece was loaded, and the recovery time (Tf) of the sensor was 8 ms when the weight piece was unloaded. It was also found through experiments that, when a 50 mg weight piece was loaded on the sensor (with pressure approximately 6.8 Pa), the capacitance output of the sensor had a recognizable increase change, and the sensor could effectively maintain the change in the capacitance value. The response curve of the sensor pressure limit detection is shown in Figure 8c. The ratio of the capacitance change to the sensor’s initial value was approximately 0.015. Unfortunately, when the mass of the weight was less than 50 mg, the capacitance value change of the sensor was difficult to recognize owing to noise interference. Therefore, 6.8 Pa can be regarded as the minimum pressure resolution that the sensor can recognize. This also proves that the sensor has certain advantages in detecting extremely low pressure. To prove that the flexible pressure sensor can adapt to complex non-planar environments by virtue of its own flexibility, the sensor was fixed on the two splints of the flexible electronic tester. One of the splints was moved inward to make the sensor bend and arch, forming different angles. The test results are shown in Figure 8d. As the bending angle increased, the capacitance of the sensor gradually increased and the speed gradually increased. When the sensor was bent, the dielectric layer and the microstructure of the electrode surface were squeezed, thereby changing the initial capacitance value. To demonstrate the capability of the sensor responding to the pressure changes even in the bent state, the sensor was attached to the outer wall of 100 mL, 50 mL, and 10 mL graduated cylinders. At this time, the bending angles of the sensor were approximately 38°, 45°, and 76°, respectively. The sensitivity tests were performed on the sensors in the three bending states (Figure 8e). The results indicate that, compared with the sensitivity curve of the sensor in the flat state, the sensitivity of the sensor in the bent state is obviously lower. This is because the sensor was compressed when the sensor was bent, which increased the initial capacitance value of the sensor; that is, $C_{0}$ in $\Delta C/C_{0}$ increased, resulting in a decreased slope of the pressure–capacitance curve and a decrease in sensitivity. In the sensor bending test, the pressure response curve when the sensor was bent at 36° was the steepest, followed by the pressure response curve at 45°, and the pressure response curve at 76° was the flattest. This indicates that the larger the sensor bending angle, the higher is the initial capacitance value and the lower is the sensitivity of the sensor. To explore the sensor’s ability to detect different pressure loading signals, sharp pulse pressure and square wave pressure of signal values 0.3, 0.8, and 1.5 kPa were repeatedly applied to the sensor twice. As shown in Figure 8f,g, under the same applied pressure, irrespective of whether square wave pressure or sharp pulse pressure was applied, the corresponding capacitance output response value was almost the same. This indicates that the sensor has a stable response to different types of pressure signals. To analyze whether the sensor could work normally for a long period while generating heat or in a high-temperature environment, the sensor was placed in a vacuum drying oven and the temperature increased from 20 °C to 80 °C. Then, the capacitance change curve of the sensor during the heating process was measured, and the results are as shown in Figure 8h. It can be seen that the capacitance change of the sensor from 20 °C to 80 °C was only 3.1% of the initial capacitance value. Thus, the impact on the electrical performance of the sensor is small and the sensor meets the working requirements under high temperature.

To show the durability and service life of the sensor, a cyclic pressurization–unloading test was performed on the sensor. The test pressure was 200 and 400 Pa, and the number of repetitions was approximately 2000 (Figure 9). As shown in Figure 9a,d, the capacitance response curves can maintain good stability at 200 and 400 Pa, indicating that the sensor has good repeatability under different pressure. As shown in Figure 9b,c and Figure 9e,f, at different time periods of 1000–1040 s and 4000–4040 s, the capacitance response curves are very close, and there is no consistent increase or decrease in the capacitance change. This proves that the sensor has good durability and creep resistance. 

In the repeatability test with pressure 200 and 400 Pa, a loading–unloading process was carried out to explore the hysteresis of the sensor under dynamic pressure. As shown in Figure 10a,b, the hysteresis of the sensor under a dynamic pressure of 200 Pa was approximately 5.9% and that under a dynamic pressure of 400 Pa was approximately 9%. The sensor has suitably low hysteresis. To explore the sensitivity change of the sensor after a long-term operation, the sensitivity of the sensors under repeated loading–unloading cycles for a total of more than 10,000 times was tested and compared with the initial sensitivity (Figure 10c). The test results show that, after more than 10,000 pressure loading–unloading cycles, the sensitivity curve of the sensor basically fit the initial sensitivity curve, and the maximum error was approximately 8%. However, the sensitivity of the sensor after 10,000 repetitions was slightly higher than the initial sensitivity. This may be due to the performance abnormality caused by fatigue damage to the flexible material of the sensor under long-term loading.

The performance of the proposed sensor is compared with that of some reported sensors (Table 1). The proposed sensor evidently has certain advantages in terms of sensitivity, minimum resolution, response time, and stability.

### 3.2. Acquisition and Denoising of Sensor Signals

Under the premise of not affecting normal human activities, a portable and wearable hardware circuit board was designed to flexibly collect three-channel capacitive sensing signals in real time. Using the built-in Bluetooth system of the microcontroller module, the collected signals were transmitted to the PC by wireless transmission to complete the storage and analysis of the sensor data. The actual image and installation position of the hardware circuit board are shown in Figure 11. The hardware circuit includes the capacitance acquisition module (PCap02, German ACAM company, Stutensee, Germany), microcontroller module (CC2541, TI company, Dallas, TA, USA), switch module (ADG711, Analog Devices Inc., Norwood, MA, USA), and power module (TLV75733PDBVR, TI company). The power supply voltage of the circuit board was 3.3 V.

The multi-channel sensor signal acquisition system was used to realize the data acquisition of respiratory signals, elbow joint bending signals, and knee joint bending signals. The acquisition results are shown in Figure 12. Figure 12a shows that the sensor responds more accurately to the three breathing states, namely normal breathing, deep breathing, and rapid breathing. The intensity of deep breathing is greater than that of normal breathing, while the frequency of rapid breathing is higher. In addition, the intensity of inhalation during rapid breathing is similar to that of deep breathing, but the exhalation is not thorough enough. The real-time acquisition of the change in the sensor capacitance has a better mapping ability with the actual breathing situation. Figure 12b,c, respectively, show the capacitance response of the sensor to the bending state of the elbow joint and the knee joint. Between them, the elbow joint is a smaller joint, and the sensor has a larger bend when the elbow joint is bent, and thus can produce a larger change in capacitance. Because the knee joint is a larger joint, even if the knee joint is bent at a larger angle, owing to the small size of the sensor, its capacitance response is slightly lower than that when the elbow joint is bent. The test curve results also prove that the sensor can generate effective responses to different joint motion signals.

Therefore, in this study, three sensors to collect breathing signals, elbow joint bending signals, and knee joint bending signals were installed to distinguish the four typical motion states of the human body, namely sitting, standing, walking, and running. The process is as follows. First, the three flexible sensors were attached to the volunteer’s chest, elbow, and knee joints and attached to a hardware circuit board. The volunteer maintained the four states for 1 min each and repeated them 15 times. Considering a breathing frequency range of 0.15–0.4 Hz and taking into account the movement frequency of the elbow and knee joints, a sensor signal acquisition frequency of 5 Hz was set in this study to avoid introducing excessive noise. The acquired sensor data are shown in Figure 13a–d. 

As indicated by the black curve in Figure 14, the acquired signals inevitably comprise various noise signals because of the limitations of environment and experimental conditions during the signal acquisition process. Herein, to denoise the collected sensor signals, the wavelet threshold denoising method was adopted. The parameter setting rules of the wavelet threshold denoising method are as follows: the wavelet basis function is coif4, the number of wavelet decomposition layers is 5, the rigrsure threshold rule is used as the threshold method, and the soft threshold function is used as the threshold function. The data of different motion states collected by the sensor were transmitted to PC through Bluetooth for storage and denoising. The red curve in Figure 14 shows the denoising effect of partial breathing signals collected in the state of standing still. Clearly, the noise in the signals was successfully removed, and the signals after denoising had better smoothness and better retained the original change trend and characteristics of the signals.

### 3.3. Construction of BP Neural Model and Testing of Human Motion State Recognition System

In this study, windowing processing was conducted on the collected signals; that is, the relatively long data were divided into fixed-length data fragments as samples. The window length was set to 10 s, and the signals were divided into a series of analysis samples with a sliding window in 5 s increments, thereby generating a 5 s overlap. The signal overlap method was used to maximize the use of continuous data flow. The signal sampling frequency of the sensor was 5 Hz; that is, every 50 data points were regarded as a sample. A sample was collected every 25 data points, thus resulting in a total of 716 sets of samples in the four motion states. Each set of samples includes respiration signals, elbow joint bending signals, and knee joint bending signals with a 10 s length. The sample features were extracted using a time-domain feature extraction method. Five common time-domain features were selected, namely the maximum value, minimum value, average value, root mean square, and variance. A set of samples included three sets of data collected by the sensor—breathing signals, elbow joint bending signals, and knee joint bending signals, denoted as A, B, and C, respectively. Through calculation, the eigenvalues of the five time-domain features of the three sets of data were obtained. A sample with 15 characteristic parameters was formed as the input for the neural network training. Subsequently, a three-layer forward BP network was established, including an input layer, a hidden layer, and an output layer. The input layer of the BP network had 15 nodes. The 15-dimensional input vectors formed sequentially corresponded to the 15 eigenvalues in a set of samples. The output layer had a node which represented the motion states of sitting, standing still, walking, and running with outputs 1, 2, 3, and 4, respectively. The number of neurons in the hidden layer was seven. From the obtained 716 groups of sample sets for distinguishing the four motion states, 70% of the samples were randomly selected as the training set, 15% of the samples as the validation set, and 15% of the samples as the test set. The Levenberg–Marguardt algorithm was adopted as the training algorithm.

The mean square errors of the three sample sets after 26 training iterations are shown in Figure 15. The verification set reached the optimal mean square error of 0.13724 at the 20th iteration. At this time, the mean square errors of the training set and the test set were 0.10342 and 0.11179, respectively. The lower the mean square error, the better is the effect of the neural network. Finally, the human body motion state recognition system was established (Figure 16). A new type of flexible capacitive sensor was applied for sensing human physiological signals. The data were transmitted to the PC through the signal acquisition system. After denoising, sample partition, and feature extraction, the processed data were used for neural network training, and the trained neural network model was used to perform human motion state recognition. Through the application of the signal acquisition system, a total of 100 test samples were obtained, including 25 each of sitting, standing, walking, and running samples. These samples were used to test the reliability of the trained human motion state recognition network. The test results obtained are shown in Table 2.

As shown in Table 2, the overall correct recognition rate of the test was 94%. In detail, the highest recognition rate was obtained for sitting and standing still, reaching 96%, followed by walking and running, which were both 92%. The first two states have extremely low motion amplitudes with only slight unconscious motion of the joints. Hence, the data collected by the sensor were relatively stable. By contrast, because the motion amplitudes of walking and running are large, the signals collected by the sensor changed drastically. The training samples are not sufficiently comprehensive to cover all possible samples.

## 4. Discussion and Conclusions

In summary, this paper presented a method for human body motion state recognition based on a flexible, wearable capacitive pressure sensor. The sensor used sandpaper as a template, MWCNTs as a conductive material, and PDMS as a flexible substrate to obtain a MWCNTs/PDMS electrode layer with microstructure embedded on the surface. To reduce the interference of parasitic capacitance, the dielectric layer was made of a BaTiO_3_/PDMS composite film with a mass ratio of 30%, which effectively improved the dielectric constant of the sensor. The designed flexible pressure sensor showed a sensitivity up to 2.39 kPa^−1^ in a pressure measurement range of 0–120 kPa, a minimum pressure resolution of 6.8 Pa, fast response time (16 ms), and fast recovery time (8 ms). In the dynamic response, the designed sensor had a low hysteresis and could remain stable after repeated loading over 10,000 times. The hardware circuit board was used to collect the breathing signals, the elbow joint bending signals, and the knee joint bending signals of the human body in different motion states. The collected sensor signals were denoised by the wavelet threshold denoising method. Subsequently, the denoised data were divided into samples. The time-domain feature values were extracted to form feature samples. The BP neural network was used to train and learn the samples to obtain a motion state analysis model. Finally, the human motion state recognition system was constructed, which realized the recognition of four motion states, namely sitting, standing, walking, and running. After testing, the comprehensive correct recognition rate of the system was approximately 94%. This method provides a new concept for the field of human motion recognition and has great application potential in human–computer interaction and health monitoring.

## Figures and Tables

**Figure 1 micromachines-12-01219-f001:**
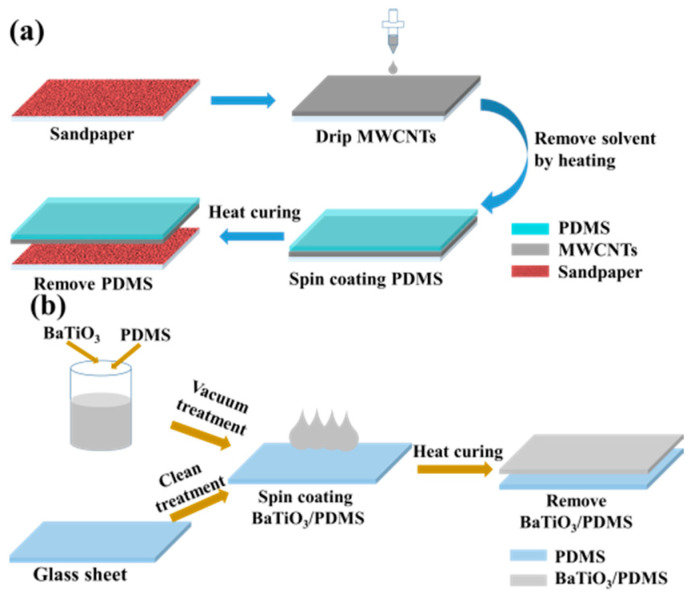
Fabrication process of the sensor. Fabrication process of the (**a**) microstructured electrode and (**b**) BaTiO_3_/PDMS dielectric layer.

**Figure 2 micromachines-12-01219-f002:**
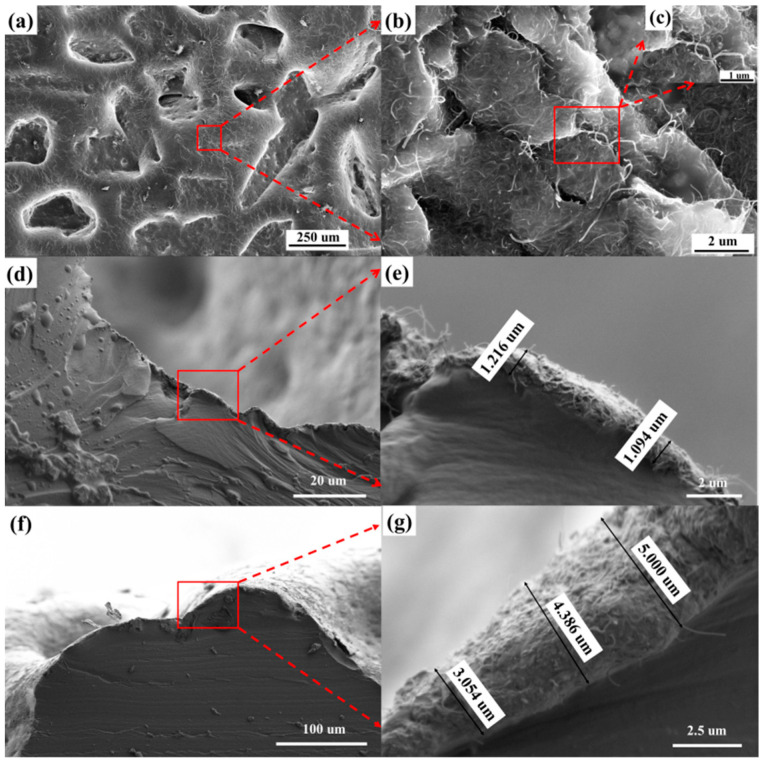
SEM image of the electrode layer. (**a**) The surface of electrode layer. (**b**) The MWCNTs layer. (**c**) Further amplification of the MWCNTs layer. (**d**) The cross-section of the bottom of the microstructure groove. (**e**) The thickness of the MWCNTs layer at the bottom of the microstructure groove. (**f**) The cross-section on the top of the microstructure bulge. (**g**) The thickness of the MWCNTs layer on the top of the microstructure bulge.

**Figure 3 micromachines-12-01219-f003:**
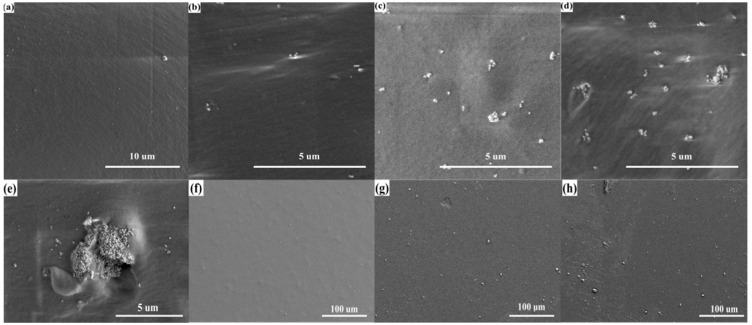
SEM images of the dielectric layer. (**a**) The cross-section of pure PDMS film; and (**b**–**d**) composite film with BaTiO_3_ content of 10%, 20%, and 30%, respectively. (**e**) Agglomeration of BaTiO3 particles. (**f**–**h**) The surface of composite film with BaTiO_3_ content of 10%, 20%, and 30%.

**Figure 4 micromachines-12-01219-f004:**
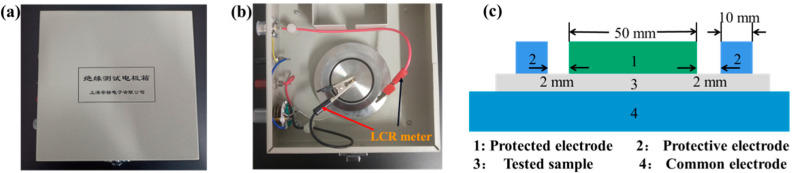
The three-electrode measurement system. (**a**) Exterior diagram of the test system. (**b**) Interior diagram of the test system. (**c**) Sectional diagram.

**Figure 5 micromachines-12-01219-f005:**
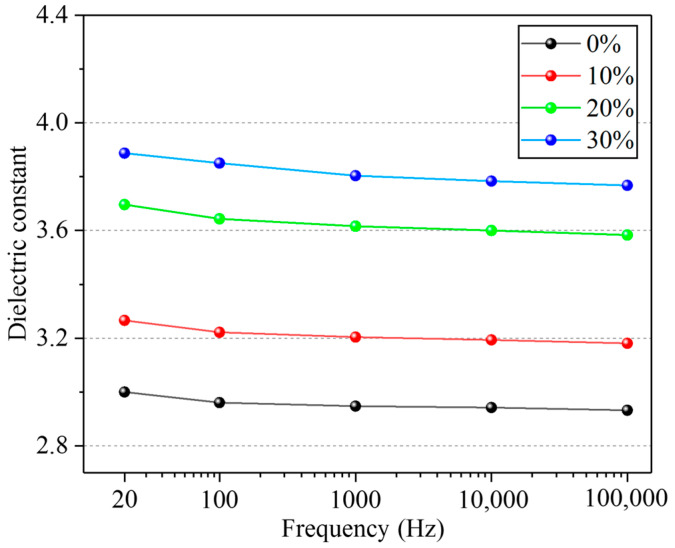
Relative dielectric constant of PDMS composites with BaTiO_3_ content of 0%, 10%, 20%, and 30%.

**Figure 6 micromachines-12-01219-f006:**
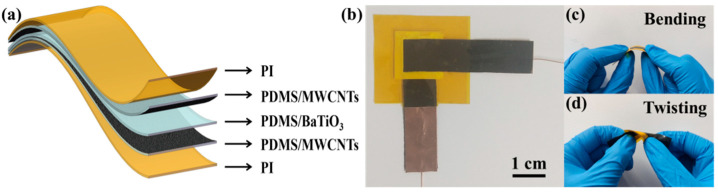
Sensor packaging. (**a**) Structure diagram; (**b**) sensor image; (**c**) the bending state of the sensor; (**d**) the twisting state of the sensor.

**Figure 7 micromachines-12-01219-f007:**
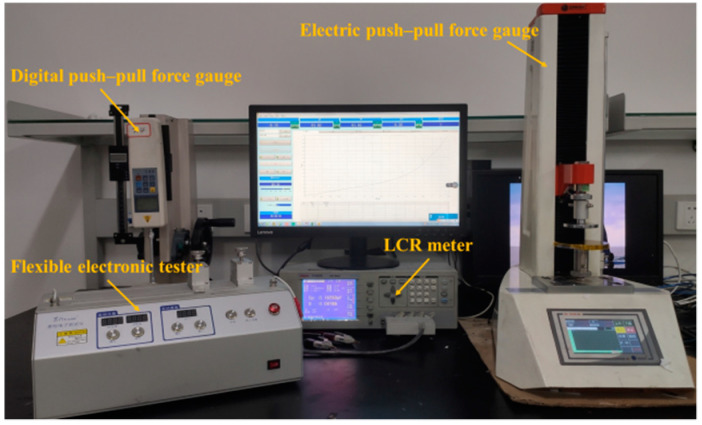
Performance test device of sensor.

**Figure 8 micromachines-12-01219-f008:**
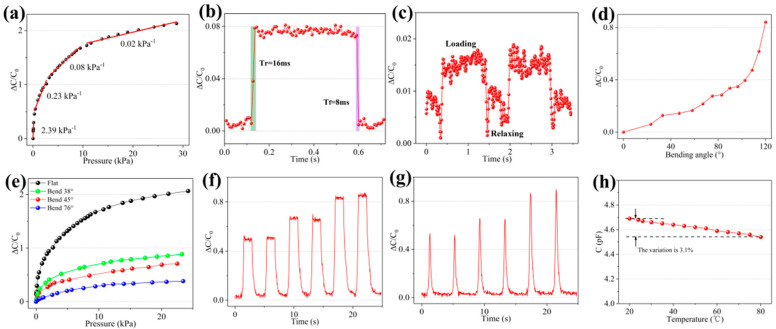
Performance test curve of the sensor. (**a**) Sensitivity. (**b**) Response time and recovery time. (**c**) Limit of pressure sensing. (**d**) Output response of the sensor in the bending state. (**e**) The sensitivity of the sensor in the bending state. (**f**) Detection capability under square wave loading. (**g**) Detection capability under sharp pulse loading. (**h**) Effect of temperature on the sensor performance.

**Figure 9 micromachines-12-01219-f009:**
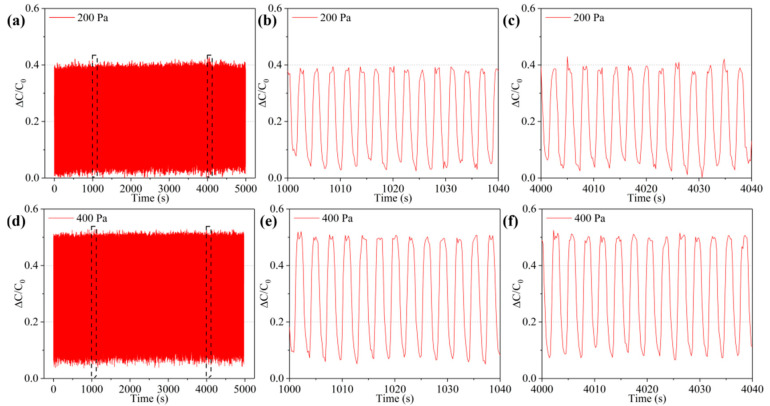
Repeatability test of the sensor. (**a**) Capacitance cyclic strain curve of the sensor under a pressure of 200 Pa; (**b**,**c**) partial graphs of Figure 6a; (**d**) capacitance cyclic strain curve of the sensor under a pressure of 400 Pa; (**e**,**f**) partial graphs of Figure 6d.

**Figure 10 micromachines-12-01219-f010:**
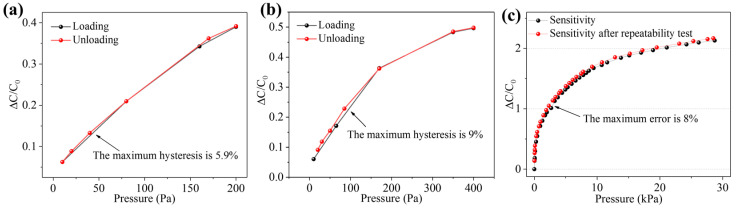
Sensor hysteresis and the sensitivity curve after repeatability test. (**a**) The hysteresis of the sensor under a dynamic pressure of 200 Pa. (**b**) The hysteresis of the sensor under a dynamic pressure of 400 Pa. (**c**) The sensitivity change of the sensor after a long-term repeated loading–unloading process.

**Figure 11 micromachines-12-01219-f011:**
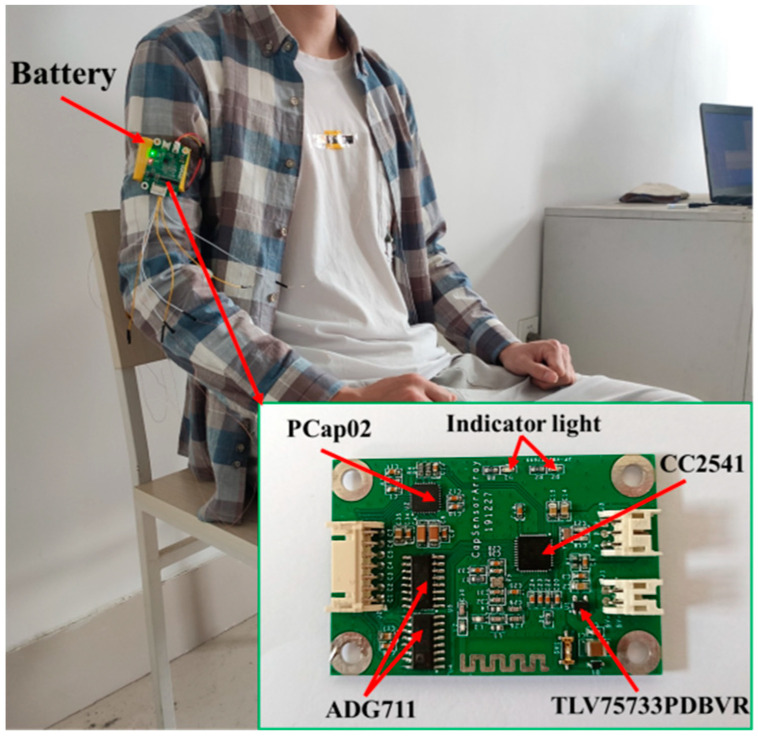
The actual image and installation position of the hardware circuit board.

**Figure 12 micromachines-12-01219-f012:**
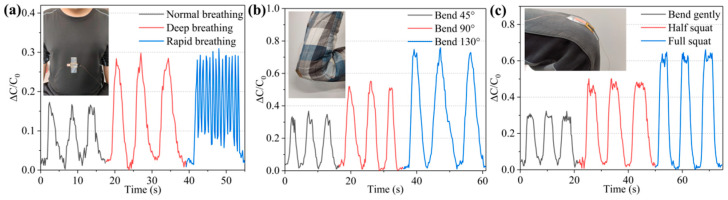
Human body signals acquired by the sensor. (**a**) Different breathing states. (**b**) Different degrees of elbow joint bending. (**c**) Different degrees of knee joint bending.

**Figure 13 micromachines-12-01219-f013:**
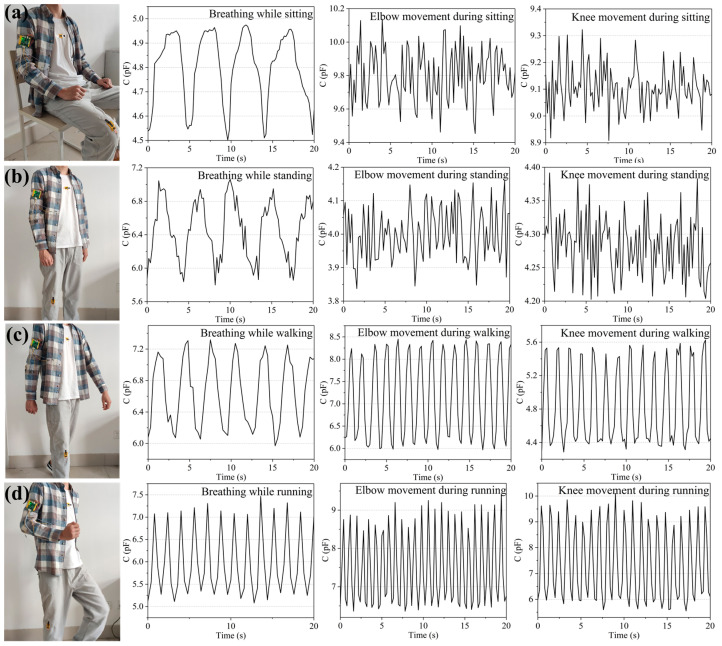
Capacitance response of the sensor in different motion states. The sensor capacitance changes in the state of (**a**) sitting, (**b**) standing still, (**c**) walking, and (**d**) running.

**Figure 14 micromachines-12-01219-f014:**
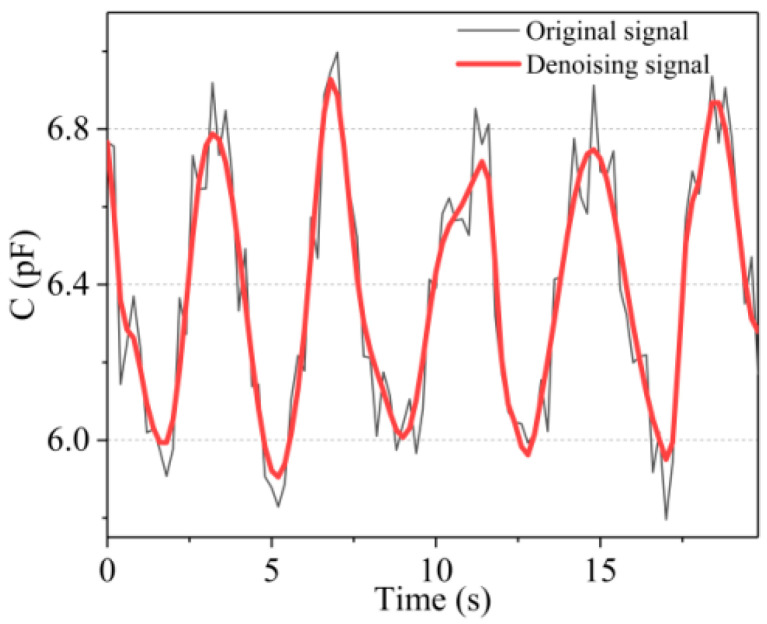
Denoising effect of wavelet threshold denoising method on breathing signals collected in the state of standing still.

**Figure 15 micromachines-12-01219-f015:**
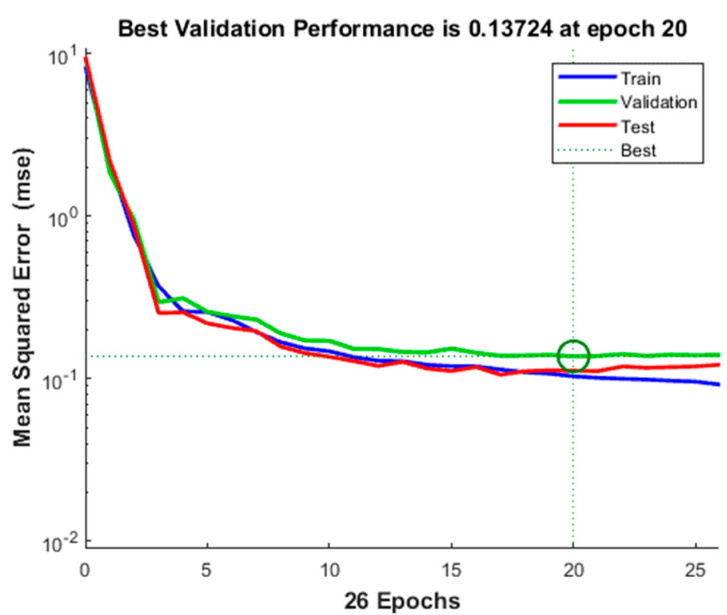
Mean square error training results of training set, validation set, and test set.

**Figure 16 micromachines-12-01219-f016:**
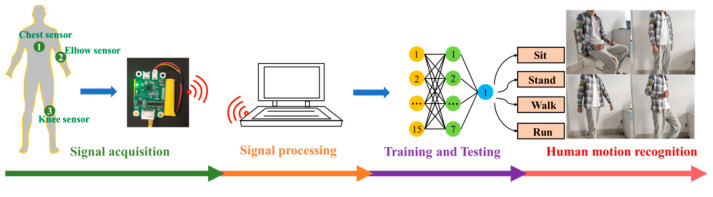
Human motion state recognition system.

**Table 1 micromachines-12-01219-t001:** Performance comparison between the proposed sensor and other sensors reported in the literature.

Sensor Structure Characteristics	Sensitivity (kPa^−1^)	Optimal Sensitivity Range (kPa)	Minimum Resolution (Pa)	Response Time (s)	Stability (Cycles)	Reference
PDMS film with pyramidal microstructure	0.55	0–0.2	3	<1	thousands of times	[26]
PDMS film with wave-shaped microstructure	4.9	0–2.5	<1.7	<0.05	5000	[31]
Dielectric layer with electrospinningcomposite fiber film	0.99	0–1.2	~	0.029	1000	[32]
Electrode layer with icicle-shaped liquid metal film	0.39	0–1	12	0.19	6000	[33]
Microstructured PDMS film coated with reduced graphene oxide	25.1	0–2.6	16	0.12	3000	[34]
PDMS film with sandpaper microstructure	0.3954	0–2.67	~	0.49	6000	[35]
Microstructured PDMS spraying Ag nanowells	0.2837	0–1.3	300	0.05	~	[36]
Microstructured PDMS surface embedded with MWCNTs	2.39	0–0.12	6.8	0.016	>10000	This work

**Table 2 micromachines-12-01219-t002:** Test results of the neural network model.

Test Samples	Four Outputs of BP Neural Network				Correct Recognition Rate (%)
	Sitting	Standing	Walking	Running	
Sitting	24	1	0	0	96
Standing still	1	24	0	0	96
Walking	1	1	23	0	92
Running	0	0	2	23	92

## Data Availability

Not applicable.

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
