# Peer review of "Human Motion State Recognition Based on Flexible, Wearable Capacitive Pressure Sensors"

_micromachines, 2021, doi:10.3390/mi12101219_

Round 1

Reviewer 1 Report

The authors introduced not only an innovation of flexible micro-patterned pressure sensors based on micro-structured MWCNTs/PDMS electrodes and BaTiO3/PDMS dielectric layer but also a demonstration of using their devices for the human motion recognition. In general, the concept and scope of this study are quite good. However, there exists some imperfect explanations in this manuscript. Therefore, the reviewer suggests a minor revision for this.

Comment 1. In Eq. (3), P is considered as the amount of pressure exerted on the sensor, instead of “the change in pressure” (delta P).

Comment 2. The magnification of SEM images in Fig. 2 are not suitable to observe and evaluate the agglomeration of BaTiO3 particles with a size of 500 nm. The authors should show other images having higher magnifications. Besides, please provide an optical or electronic image of micro-structured MWCNTs/PDMS electrodes.

Comment 3. In the section 2.4 (line 202), what is the thickness of the electrode and dielectric layers employed in the sensor devices? Perhaps, the thickness of the dielectric layer is quite thick, leading to the small capacitance value of the device (< 7 pF) as shown in Fig. 11. So, if the capacitance value of the device increased like 100 pF, would it improve the device response and sensitivity?

Comment 4. In line 230-232, the explanation is fine for explaining how “deltaC/C0” still increased when extensive pressure exerted on the sensor. However, it is not reasonable for interpreting the nonlinear behavior of the sensor’s sensitivity. At high pressurization, the saturated response causing the reduced sensitivity of the stretchable PDMS-based composites should be considered in terms of “elastic regime” and “plastic regime” (in stress-strain curves). The author can perform tensile tests to determine these two regimes of the composites.

Comment 5. In Fig. 5f,g, the response time and recovery of the device are about 1 s. Why are they so much longer than Tr and Tf (16 and 8 ms, respectively) mentioned in Fig. 5b?

Reviewer 2 Report

The manuscript "Human Motion State Recognition Based on Flexible, Wearable Capacitive Pressure Sensors" introduces a new type of flexible pressure sensor connected to a customized PCB data acquisition board. The sensor shows high sensitivity in an extensive application range, which is impressive. Plus, the authors introduced a designed neural network algorithm to denoise the collected data. The work is new and comprehensive, thus demonstrates promising future application in wearables. On the other hand, the manuscript needs to provide more explanations and convincing results before the reviewer can recommend this manuscript to be published. The authors are recommended to address the following questions and improve the quality of the manuscript.

  1. On Line 14, please justify the benefit of adding barium titanate? If the high dielectric constant of barium titanate is the consideration, why not using other high-dielectric constant materials, such as PZT or some polymers? Please also justify the reason for selecting these three doping concentrations.

  1. In the Experimental Section, please include elaborations on sensor testing methods, such as sensor performance test, signal denoising, reliability test, etc.

  1. On Line 149, how thick is the MWCNT layer? It would be better to provide an SEM image and show the morphology of MWCNT layer. What is the advantage of using MWCNT?

  1. On Line 156, this is a smart idea to make use of the rough surface of the sandpaper, but the process is not well controllable or repeatable. How would the non-uniform surface affect the performance of the device since the air gap between protrusions of the PDMS surface is quite random? Please also justify the reason for using 80-mesh sandpaper but not higher or lower grit ones.

  1. In Figure 2, please show a larger area SEM of each film with different particle concentrations. It can show the distribution uniformity of the particles in the film. The uniformity would affect the device's performance.

  1. On Line 186, an image of the measurement setup or detailed descriptions of the measurement would be helpful in better understanding how these experiments were carried out.

  1. On Lines 191­194, what is the purpose of measuring dielectric constant under high frequency? If the changing direction of the dipole is the reason for the declined dielectric constant, it would be interesting to see the dielectric constant value under a lower than 20 Hz regime.

  1. On Line 205, what is the total thickness of the device? A close look photo of the device under deformation such as bending, twisting, and stretching would be helpful.
  2. On Line 246, what specific noise is discussed here? Understanding the noise source would help in finding proper approaches to denoise or denoising algorithm optimization.

  1. In Figure 5 (b), the sampling rate in this measurement is not high enough to obtain an accurate response time. The actual response time could be smaller than the results shown here.

  1. In Figure 10, it would make a better demonstration when you place the device directly on the human skin surface rather than on clothes. What about the measurement on smaller joints, such as finger joints? The device would bear greater stain on smaller joints, which is more challenging so more interesting.

  1. In Figure 11, why would the sensor's capacitances under four states have significant discrepancies? In other words, where are these discrepancies from?

  1. On Line 371, the denoising method needs more explanation. Is it on-site processing or post-processing?

Reviewer 3 Report

In this manuscript, the authors describe a method for human body motion state recognition based on a flexible, wearable capacitive pressure sensor. The capacitive sensor with BaTiO3/PDMS composite film as the dielectric layer has the characteristics of microstructure, wide detection range and good stability. Thus, I would recommend the acceptance of this manuscript for Micromachines with major revision.

  1. The innovation point of the article is not highlighted. If the innovation point is to use neural network training in sensing data identification, the advantages of this method should be introduced in Introduction.
  2. Why did the author choose BaTiO3 as the dielectric material? What are the advantages and characteristics of this material that should be introduced in this paper?
  3. BP neural network identification of data is mainly based on the change of capacitance, so whether the same capacitance change will mislead the identification of such behavior.

Round 2

Reviewer 3 Report

The revised manuscript is acceptable.